# Sensitive Detection of Cell-Free Tumour DNA Using Optimised Targeted Sequencing Can Predict Prognosis in Gastro-Oesophageal Cancer

**DOI:** 10.3390/cancers15041160

**Published:** 2023-02-11

**Authors:** Karin Wallander, Zahra Haider, Ashwini Jeggari, Hassan Foroughi-Asl, Anna Gellerbring, Anna Lyander, Athithyan Chozhan, Ollanta Cuba Gyllensten, Moa Hägglund, Valtteri Wirta, Magnus Nordenskjöld, Mats Lindblad, Emma Tham

**Affiliations:** 1Department of Molecular Medicine and Surgery, Karolinska Institutet, 17176 Stockholm, Sweden; 2Department of Clinical Genetics, Karolinska University Hospital, 17164 Stockholm, Sweden; 3Science for Life Laboratory, Department of Microbiology, Tumor and Cell Biology, Karolinska Institutet, 17165 Stockholm, Sweden; 4Science for Life Laboratory, School of Chemistry, Biotechnology and Health, Royal Institute of Technology, 10044 Stockholm, Sweden; 5Genomic Medicine Center Karolinska, Karolinska University Hospital, 17164 Stockholm, Sweden; 6Department of Clinical Science, Intervention and Technology, Karolinska Institutet, 14152 Huddinge, Sweden; 7Department of Upper Abdominal Diseases, Karolinska University Hospital, 17164 Stockholm, Sweden

**Keywords:** gastric, oesophageal, cancer, liquid biopsy, cell-free (tumour) DNA, prognostic biomarker

## Abstract

**Simple Summary:**

Cancer in the stomach and oesophagus is deadly when discovered at a late stage. There are no good biomarkers for its detection or for making a prognostic prediction. In this study, we evaluate the analysis of cell-free DNA as a prognostic cancer biomarker. Cell-free DNA is DNA released from any tissue to a body fluid. When there is a tumour in the body, some of the cell-free DNA will come from that tumour, and it can be detected in a blood sample. We show that the detection of cell-free DNA from the cancer correlates to a worse prognosis than when no tumour DNA is detected. We also show that the method of analysis is important. Either a tissue biopsy must be included as a validation of the genetic variants detected or analysis of the blood cells or another blood sample after tumour resection needs to be analysed to improve detection.

**Abstract:**

In this longitudinal study, cell-free tumour DNA (a liquid biopsy) from plasma was explored as a prognostic biomarker for gastro-oesophageal cancer. Both tumour-informed and tumour-agnostic approaches for plasma variant filtering were evaluated in 47 participants. This was possible through sequencing of DNA from tissue biopsies from all participants and cell-free DNA from plasma sampled before and after surgery (n = 42), as well as DNA from white blood cells (n = 21) using a custom gene panel with and without unique molecular identifiers (UMIs). A subset of the plasma samples (n = 12) was also assayed with targeted droplet digital PCR (ddPCR). In 17/31 (55%) diagnostic plasma samples, tissue-verified cancer-associated variants could be detected by the gene panel. In the tumour-agnostic approach, 26 participants (59%) had cancer-associated variants, and UMIs were necessary to filter the true variants from the technical artefacts. Additionally, clonal haematopoietic variants could be excluded using the matched white blood cells or follow-up plasma samples. ddPCR detected its targets in 10/12 (83%) and provided an ultra-sensitive method for follow-up. Detectable cancer-associated variants in plasma correlated to a shorter overall survival and shorter time to progression, with a significant correlation for the tumour-informed approaches. In summary, liquid biopsy gene panel sequencing using a tumour-agnostic approach can be applied to all patients regardless of the presence of a tissue biopsy, although this requires UMIs and the exclusion of clonal haematopoietic variants. However, if sequencing data from tumour biopsies are available, a tumour-informed approach improves the value of cell-free tumour DNA as a negative prognostic biomarker in gastro-oesophageal cancer patients.

## 1. Introduction

Gastro-oesophageal cancer is common and deadly. The symptoms are diffuse and usually occur late; thus, the cancer is often advanced at the time of diagnosis. For localised diseases, surgical resection, often in combination with chemotherapy, may be curative. However, most patients have advanced disease at the time of diagnosis or experience an early relapse [1]. To improve the prognosis, there is a need for a biomarker that is highly specific for gastro-oesophageal cancer, enabling earlier diagnosis, optimal individual cancer treatment, the early detection of residual or recurrent disease, and evaluation of the given therapy.

Cell-free tumour DNA (ctDNA) is circulating cell-free DNA released from the tumour into the bloodstream and other body fluids and has been suggested both as a diagnostic and a prognostic cancer biomarker [2,3]. There are only a limited number of studies on ctDNA in individuals with gastro-oesophageal cancer, and the cohorts are generally small. These studies suggest that high levels of ctDNA at diagnosis correlate with a poor prognosis [4,5].

We previously performed a proof-of-principle study, wherein we analysed copy number aberrations (CNAs) in ctDNA isolated from plasma collected from individuals with gastro-oesophageal cancer [6]. In the present study, we aim to analyse single-nucleotide variants (SNVs) in ctDNA from plasma sampled before or after surgery as a prognostic biomarker in gastro-oesophageal adenocarcinoma. We hypothesised that the prognosis, and thus survival, would be worse if ctDNA was detectable than when it was not. To this end, we use a gastro-oesophageal cancer-specific, custom-designed gene panel targeting 30 genes. We compare tumour-informed and tumour-agnostic approaches for the variant analysis, as well as assess the relevance of using UMI error correction approaches for sensitive variant detection in plasma.

## 2. Materials and Methods

### 2.1. Study Design

A prospective observational cohort study was performed on patients with a newly diagnosed and resectable gastro-oesophageal adenocarcinoma. No stratification or matching was used. Cancer tissue and blood samples were collected for DNA isolation. Single nucleotide variants (SNVs) in cell-free DNA were analysed by two different analysis approaches: tumour-informed (i.e., with tissue DNA available) and tumour-agnostic, as well as two types of bioinformatic algorithms. The addition of white blood cells (WBC) and/or multiple samples to safely exclude clonal hematopoietic variants was also evaluated. This way, we could compare and evaluate variant filtering methods to determine a robust protocol for future clinical implementation. The overall study approach is schematically depicted in Figure 1.

### 2.2. Inclusion

All patients diagnosed with resectable gastric or oesophageal adenocarcinoma referred to the Department of Upper Abdominal Diseases at Karolinska University Hospital, Stockholm, Sweden, from September 2016 and February 2020 were prospectively invited to participate in the study, see Appendix A for participant flow diagram. The study was approved by the Regional Ethical Review Board in Stockholm (registration number 2016/2-31/1, with amendments 2016/1689-32, 2017/599-32/1, 2018/1472-32/1, 2019/01222). All participants gave, before inclusion, written informed consent for participation in the study and for publication of the results.

Individuals with at least one plasma sample obtained before any treatment (gastrectomy, esophagectomy, chemotherapy, or radiotherapy), and at least one sample obtained after surgical resection were included. All included individuals had adenocarcinomas and an available tumour tissue sample. For all individuals, there was a pathology report issued from the Pathology Department, Karolinska University Hospital, and a multidisciplinary decision of the clinical tumour stage, reported as TNM (tumour, lymph node, and metastasis stage). The highest CRP (C-reactive protein) concentration value during inpatient care after surgical treatment was noted (CRPmax).

The sex distribution and age at diagnosis in our cohort is comparable to all gastro- and oesophageal patients in Sweden, with 36% versus 30% women respectively, and average age at diagnosis of 71 years in both cohorts [7]. Although the eligible patients were thought to be resectable at inclusion, eight were proved to be metastatic either during surgery or retrospectively determined probably metastatic already at surgery. All participants were followed until death or until the end of the study. The median follow-up time was 28 months (3-65).

### 2.3. Tissue DNA Isolation

Fresh tumour tissue biopsies were obtained by the surgeon either at the time of diagnostic gastroscopy or at surgical resection (in most cases total gastrectomy). One sample was sent for histopathology and a separate sample was used in this study; frozen within one day and later used for DNA extraction as previously described [6], according the manufacturers’ instructions for EZ1 DNA Tissue Kit (Qiagen, Hilden, Germany). 250 ng DNA was used from each sample and DNA was frozen at −20 °C.

### 2.4. Plasma DNA Isolation

Blood samples were collected in Streck^®^ tubes (La Vista, NE, USA), and plasma and cell-free DNA were isolated as previously described [6]. In summary, the blood cells were removed by double centrifugation within 5 days (10 min at 4 °C 1600× *g* and then 10 min at 16,000× *g*). Plasma was stored at −80 °C. Cell-free DNA was extracted from 2–5 mL of plasma with the QIAamp DNA Blood Mini Kit (Qiagen, Hilden, Germany) with washes performed in Qiacube (Qiagen, Hilden, Germany). The cell free DNA was eluted in 40 microliters of nuclease free water and stored at −20 °C without being defrosted before library preparation. Cell-free DNA concentration was measured by Qubit (Thermo Fisher, Waltham, MA, USA) and quality control was performed by BioAnalyzer 2100 Expert Hight sensitivity Assay (Agilent, Santa Clara, CA, USA) according to the producer’s instructions, at the time of cell-free DNA isolation. One positive control sample (1% ctDNA complete reference material) from SeraCare (SeraCare, Milford, MA, USA) and one plasma sample from a normal blood donor was included in the analysis pipeline.

### 2.5. White Blood Cell (WBC) DNA Isolation

DNA from WBC was either obtained from a separate EDTA test tube, when available, or from the buffy coat cells in the Streck^®^ tubes. DNA was isolated from 2–3.5 mL blood using EZ1 DNA Tissue Kit (Qiagen, Hilden, Germany) or on Qiasymphony (Qiagen, Hilden, Germany) at the Department of Clinical Genetics, Karolinska University Hospital, Stockholm, Sweden. A buffer change to nuclease-free water using AMPure XP according to the manufacturers’ instructions (Beckman Coulter, IN, USA) was performed before library preparation. 59–250 ng blood DNA was sequenced in the same pipeline as the cell-free DNA.

### 2.6. Design of the GI cfDNA Panel

The gene panel for the plasma analysis (GI cfDNA gene panel; cfDNA, cell-free DNA) was designed to include genes known to be associated to gastro-oesophageal and colorectal cancer according to the cBioPortal database (Appendix A) [8]. All genes with a high mutation frequency and with an established cancer connection were included in the panel. If there were hotspot regions, only those were included. The final panel included 30 genes (with hotspots only in 26 of the genes) and had a size of 76,261 base pairs (https://clincial-genomics-hybrid-capture.readthedocs.io/en/latest/panel_doc/03_gi_cfdna_3.1.html; accessed on 1 March 2020). The panel was expected to detect 84% of all gastro-oesophageal cancer single nucleotide variants.

### 2.7. Validation of the GI cfDNA Panel

For validating the GI cfDNA gene panel and the efficiency of the UMI workflow, we used three commercially available reference samples from SeraCare which contained 14 target mutations (SNVs and indels) (SeraCare, Milford, MA, USA) with variant allele frequencies (VAFs) 0.1%, 0.5% and 1% [Material numbers: 0710-0671 (https://www.seracare.com/Seraseq-ctDNA-Complete-Reference-Material-AF1-0710-0671/; accessed on 1 March 2020), 0710-0672 (https://www.seracare.com/Seraseq-ctDNA-Complete-Reference-Material-AF05-0710-0672/; accessed on 1 March 2020), and 0710-0673 (https://www.seracare.com/Seraseq-ctDNA-Complete-Reference-Material-AF01-0710-0673/; accessed on 1 March 2020)] as well as normal plasma from a blood donor. A total of 10 or 50 ng input cell-free DNA from each reference sample was prepared as described below in Section 2.8, however, with the following exception: PCR amplification 8 cycles in the first PCR amplification. Sequencing was done to various read depths (20 M, 40 M, 80 M read-pairs) on the GI cfDNA gene panel (as described below). In addition, various bioinformatic parameters in TNScope using Sentieon (San José, CA, USA) were tested. The results were plotted as scatterplots using the R-package ggplot2 v3.3.3 [9] (Appendix A). Using 10 ng input and 20 M read pairs and after filtering against normal plasma (which removed 75% of all variants), a median of 127 variants were detected per sample and 13 of 14 target variants were detected in the SeraCare reference samples with VAFs 0.5% and 1%. No variants could be detected in the 0.1% VAF sample. The coverage after UMI collapsing was 1000–1100x. Increasing the cell-free DNA input to 50 ng and the read depth to 40 M read-pairs increased the coverage to 2500–3000x after UMI collapsing. The total number of filtered variants was also higher (median 253) and all 14 variants at VAF 1% were detected as well as 4 in the VAF 0.1% sample, but still only 13/14 at VAF 0.5%.Of note, the missed variant was PIK3CA_3204_3205insA, an indel. It was detected but did not pass the quality filtering. By adjusting the minimal tumour LOD (limit of detection) to 4 (instead of default 6.3), an additional variant could be detected at VAF 0.1%%). Increasing the read depth to 80 M read-pairs led to an even higher background (280 variants in total) although it did enable detection of the PIK3CA indel variant at 0.5% VAF.

Based on the validation series described above, the plasma samples were sequenced aiming at 30 M read pairs and the minimum tumour log odds in the final call of variants was set to 4 instead of the default of 6.3.

We also compared to the previous study performed by our group, with an overlapping cohort of 20 gastro-oesophageal cancer patients [6]. Three out of the four samples with available ichorCNA tumour fraction approximation (https://github.com/broadinstitute/ichorCNA; accessed on 1 March 2020) had detectable ctDNA and all had a similar but not identical highest plasma VAFs detected in this study (Appendix A).

### 2.8. Library Preparation and Sequencing

Library preparation and sequencing of DNA was performed at the Clinical Genomics Stockholm, Science for Life Laboratory (Solna, Stockholm, Sweden) [10]. Cell-free DNA was sequenced within one month from DNA isolation. The entire elution volume (corresponding to 8.5 ng–250 ng of cell-free DNA) was used. For tissue DNA, 250 ng was used from all samples.

The KAPA HyperPlus library preparation kit (Roche, Basel, Switzerland) with an additional end-repair step was used for both tissue and plasma DNA, however, no fragmentation step was used for cell-free DNA. Tissue DNA was enzymatically fragmentated for 12.5 min. The protocol also included xGen Duplex Seq adapters (3–4 nucleotide unique molecular identifiers (UMI)) for ligation, using 0.55 μM if the input amount was more than 25 ng and 0.14 μM if it was less than 25 ng) (Integrated DNA Technologies, Coralville, IA, USA), and xGen Indexing primers (2 mM with unique dual indices) (Integrated DNA Technologies, Coralville, IA, USA) were used for PCR amplification. For DNA input >200 ng, 5 cycles of amplification were performed, for 50–200 ng 8 PCR cycles and for <50 ng 13 cycles). Target enrichment was performed in a multiplex fashion with a library amount of 187.5–375 ng (4–8 plex).

The libraries were hybridized to the custom designed capture probe panel GI cfDNA (Twist Bioscience, San Francisco, CA, USA) or to a custom designed GMCK gene panel of 387 genes (tissue DNA, Karolinska Institutet, MEB) together with Twist Universal Blockers, for 16 h. The GI cfDNA gene panel was custom designed to include 30 genes known to be associated to gastro-oesophageal and colorectal cancer according to the cBioPortal database (Appendix A). The genes included in the GI cfDNA panel are *ACVR2A, AKT1, APC, ARID1A, BRAF, CDH1, CDKN2A, CTNNB1, EGFR, ERBB2, FBXW7, FGFR2, GNAS, KMT2D, KRAS, KMT2C, MSH6, MYC, NF1, NRAS, PIK3CA, POLD1, POLE, PPP2R1A, PTEN, RHOA, RNF43, SMAD4, SOX9,* and *TP53*. The GMCK gene panel was designed to enable detection of clinically relevant SNVs, indels, copy number variations, fusions, MSI (microsatellite instability), and estimation of TMB (tumour mutational burden). It contains approximately 21,000 baits, covering 1.9 Mb of target. The full coding sequence of 198 genes, and hotspot regions of 149 genes are captured. Additional baits to increase CNV calling sensitivity are added for 86 genes, intronic sequences for SNV detection for 19 genes, and full gene-body sequences of 9 genes. Only the SNVs were analysed in this study. The post-capture PCR was performed with xGen Library Amp Primer (0.5 mM, Integrated DNA Technologies, Coralville, IA, USA) for 14 cycles. Quality control was performed with the Qubit dsDNA HS assay (Invitrogen, Waltham, MA, USA) and TapeStation HS D1000 assay (Agilent Technologies, Santa Clara, CA, USA). Sequencing was done on NovaSeq 6000 (Illumina, San Diego, CA, USA) using paired-end 150 bp readout, aiming at 30 M read pairs per sample. Demultiplexing was done using Illumina bcl2fastq2 Conversion Software v2.20 or v2.20.0.422 implemented on the DRAGEN server (Illumina, San Diego, CA, USA).

In total, the successfully sequenced plasma samples had a median total read number of 112 M before and 1 M after UMI collapsing. The median coverage after UMI collapsing was 1309x (range 143–4533x).

### 2.9. Variant Identification and Filtering

The sequence data analysis included application of the software BALSAMIC v7.2.5 [11], which was used to analyse each of the FASTQ files. In summary, we first quality controlled FASTQ files using FastQC v0.11.9. Adapter sequences and low-quality bases were trimmed using fastp v0.20.1 [12]. For each sample, single-nucleotide variants (SNVs) and insertions and deletions were called using VarDict version 1.6 [13]. All variants were annotated using Ensembl VEP version 94.5 [14].

For cell-free DNA, UMI tag extraction and consensus generation were performed using Sentieon tools v202010.02 [15]. The alignment of UMI extracted and consensus called reads to the human reference genome (hg19) were done by bwa-mem and samtools from Sentieon utils [15]. Consensus reads were filtered based on the number of minimum reads supporting each UMI tag group. We applied a criteria filter of min_reads 3,1,1 which means that at least three UMI tag groups should be ideally considered from both DNA strands, where a minimum of at least one UMI tag group should exist in each of the single-stranded consensus reads. The filtered consensus reads were quality controlled using Picard CollectHsMetrics v2.25.0. Results of the quality controlled steps were summarized by MultiQC v1.9 [16]. For each sample, somatic mutations were called using Sentieon, TNscope, version 202010.02 [15] with nondefault parameters for passing the final list of variants (--min_tumor_allele_frac 0.0005, --filter_t_alt_frac 0.0005, --min_init_tumor_lod 0.5, min_tumor_lod 4 and --max_error_per_read 5 --pcr_indel_model NONE). Additionally, all variants present with a maximum allele frequency of more than 0.5% in gnomAD (version 2.1.1) were excluded before uploading to Scout All variants were finally annotated using Ensembl [14]. We used vcfanno v0.3.2 8 to annotate variants for their population allele frequency from gnomAD [17]. In parallel, variants were called by VarDict version 1.6 [13], with no UMI filtering.

The software Scout (version 4.34) was used to visualise sequence variants. Variants that fulfilled the following criteria were included: exonic or splice site variants (+/−5 bp from exon), with a VAF above 5% for tissue variants, and (a) either reported in ClinVar [18] as pathogenic or likely pathogenic or (b) VAF between 5 and 98%, a gnomAD frequency [17] in any population below 0.01. Tissue variants were excluded if reported >5 times in germline in our locally collected cohort (currently 7546 individuals under investigation for genetic diseases at the Department of Clinical Genetics, Karolinska University Hospital, Stockholm, Sweden), or re-occurred in our study cohort in more than half of our samples and were not reported as recurrent in the COSMIC database of somatic variants in cancer [19].

The origin, somatic or germline, for all known pathogenic or likely pathogenic filtered variants with a VAF 35–60% was accessed. If there were other filtered variants in the same sample with a similar VAF or if the phenotype of a germline variant was not consistent with that of the participant, the variants were considered somatic. As a quality control, we could verify that all variants filtered as germline could later be verified as germline by the WBC-filtered approach. Tissue variants were considered as cancer-associated driver events if they were reported as pathogenic or likely pathogenic in ClinVar [18], or occurred in COSMIC [19] at the same position as at least 5 other reported cancer cases. Samples with at least one such variant were included in the tumour-informed approach.

### 2.10. Variant Identification and Filtering

For plasma analysis, we compared four different bioinformatic pipelines (Figure 2). The same original plasma sample cell-free DNA data was used for all approaches. The results were analysed blinded to study endpoints.

For the tumour-informed approach (box A and B in Figure 2), each variant manually classified as cancer-associated from the tissue analysis (31 participants in total) was sought for in the GI cfDNA panel sequencing data. SNV/indel (single nucleotide variants and small insertions or deletions) calls generated by VarDict [13] and TNscope (Sentieon) with unique molecular identifiers (UMIs) [20] were analysed in parallel.

In a tumour-agnostic approach (47 participants in total), all variants called with UMIs were further visualized in Scout [21] (box C in Figure 2). First, variants re-occurring in more than 10 samples were identified (based on a blacklist created by the whole set of called variants), and excluded after manual inspection in IGV (Integrated Genomics Viewer, Broad Institute, Cambridge, USA) if they did not occur as recurrent pathogenic variants in the COSMIC database of somatic variants in cancer [19]. Then, exonic or splicing region variants +/− 5 bases from exon were filtered based on number of total reads (exclusion if less than 100 reads), ClinVar reports (exclusion if benign/likely benign by at least one source and also synonymous, or benign/likely benign by multiple sources) [18], and occurrence >5 times in the local cohort of individuals under investigation for genetic diseases at the Department of Clinical Genetics, Karolinska University Hospital, Stockholm, Sweden. Post-treatment plasma variants that were not present in the pre-treatment plasma sample were kept if they were known hot-spots in the cosmic database or previously detected in the tissue sample. Variants defined as germline from the tissue analysis, with a VAF in both diagnostic and post-surgical plasma samples of 40–55%, were considered verified germline variants and were not included in the final table. The variants were filtered from artefacts using a blacklist of recurrent variants and quality measures.

In addition, the tumour-agnostic variants were also filtered to exclude potential germline and clonal haematopoietic (CH) variants from the calls (box D and E in Figure 2). This was done using WBC sequencing data if available (21 participants). Occurrence of a filtered plasma variant in >4 reads in WBC was considered proof of CH or of a germline variant (the latter with VAF of around 50%). When no WBC data was available, the variants from the diagnostic sample were compared to the variants in the post-surgical sample (n = 23). In case of a variant occurring in both samples, it was considered CH if the VAF in the post-surgical samples was above a fourth of the VAF in the diagnostic sample. This cut-off was based on a median total cell-free DNA concentration ratio before and after surgery of 3.6, i.e., a median increase of 3–4 times after surgery.

### 2.11. Plasma Droplet Digital PCR

All plasma samples from 12 participants available from the follow-up period were analysed (2–6 samples per participant, 3–5 mL plasma per sample). The median total cell-free DNA concentration after extraction was 4.25 ng/uL (range 0.17–88 ng/uL based on Qubit measurements, Thermo Fisher, Waltham, MA, USA).

Digital droplet PCR (ddPCR) analysis was performed using QX200 AutoDG Droplet Digital PCR System (Bio-Rad, Hercules, CA, USA) according to manufacturer’s protocol, with minor modifications. ddPCR mutation assays, with probes labelled with FAM or HEX fluorophores and Iowa Black FQ quencher, were obtained from Bio-Rad (Hercules, CA, USA). For some patients, dark probes were ordered from Integrated DNA Technologies (Coralville, IA, USA) as part of a technical development collaboration with Bio-Rad (Hercules, CA, USA) (Appendix A).

Each 22 μL of singleplex ddPCR reaction mix was prepared with 11 μL of DNA/cell-free DNA sample, 5.5 μL of 4X ddPCR Multiplex Supermix for Probes (no dUTP) (Bio-Rad, Hercules, CA, USA) and 1X ddPCR mutation assay containing primers/probe mix in 900 nM/250 nM ratio. After droplet generation, PCR amplification was performed in SimpliAmp™ Thermal Cycler (Thermo Fisher, Waltham, MA, USA) using the following program: 95 °C for 10 min, 40 cycles of 94 °C for 30 s and 55 °C or 60 °C for 60 s, and finally, 98 °C for 10 min followed by an infinite hold at 12 °C. The ramp rate was set at 2 °C/s for each step.

The assays were first optimised on 10 ng tumour DNA. To determine optimal annealing temperatures for clear separation of positive and negative droplets, gradient ddPCR from 55 to 60 degrees for each singleplex and later multiplex reactions were performed. For the multiplex assays, dark probes were used for one of the wild-type signals in order to achieve better separation of clusters. Optimal assay concentration combinations were determined using amplitude multiplex ddPCR to enable clear distinction of multiple targets analysed in the same channel (Appendix A). The assays targets were *TP53* p. R273H, *KRAS* p. G13D, *PIK3CA* p. H1047R, *TP53* p. R273C, *KRAS* p. G12D, *TP53* p. R248W, *RB1* p. R358 *, *TP53* p. R196 *, *APC* p. R823 *, *TP53* c.720_766del, *TP53* p. R248Q, *ERBB2* p. S310Y and *TP53* p. R158fs *12.

For plasma analysis, all extracted cell-free DNA was used for ddPCR (11 μL of cell-free DNA sample per well, with primers/probe mix in 900 nM/250 nM ratio). ddPCR was performed as singleplex reaction (1 target per channel) in 5 participants and as amplitude multiplex reaction (2 targets per channel) in 7 participants (Appendix A). Plasma samples were analysed in triplicates along with 9–12 wells with wild-type (WT) cell-free DNA from healthy blood donors for false-positive rate estimation. For verifying assay performance, tumour tissue DNA as positive control template (PTC) and non-template control (NTC) containing nuclease free water instead of cell-free DNA template, were analysed in triplicates.

### 2.12. Plasma Droplet Digital PCR Data Analysis

The ddPCR data analysis was performed in QuantaSoft^TM^ Analysis Pro software (Bio-Rad, Hercules, CA, USA) following the Rare Mutation Detection Best Practices Guidelines by the manufacturer. Thresholds to discriminate positive and negative droplets, and multiple targets in amplitude multiplex ddPCR, were set by visualizing 1D and 2D amplitude plots of control wells.

After threshold setting, merged concentration (copies/well) for each target across replicates was extracted and VAF was calculated in singleplex ddPCR reaction as:$$VAF\;of\;target\;\left(\%\right)=100\times \frac{concentration\;of\;target\;\left(FAM\right)}{concentration\;of\;target\;\left(FAM\right)+concentration\;of\;wildtype\;\left(HEX\right)}$$
and in amplitude multiplex ddPCR reaction, with wildtype assay in HEX channel for one target only, as:$$VAF\;of\;target1\;\left(\%\right)=100\times \frac{concentration\;of\;target1\;\left(FAM\right)}{concentration\;of\;target1\;\left(FAM\right)+concentration\;of\;wildtype\;\left(HEX\right)}$$
$$VAF\;of\;target2\;\left(\%\right)=100\times \frac{concentration\;of\;target2\;\left(FAM\right)}{concentration\;of\;target1\;\left(FAM\right)+concentration\;of\;wildtype\;\left(HEX\right)}$$

Amount of cell-free DNA in copies per ml plasma or haploid genomic equivalents per ml (hGe/mL) plasma were calculated as:$$cell\;free\;DNA\;copies/\mathrm{mL}=concentration\;of\;target\times \frac{Elution\;volume\;\left(\mu L\right)}{sample\;input\;volume\;\left(\mu L\right)\times plasma\;volume\;\left(\mathrm{mL}\right)}$$

The false positive rate for each assay was calculated as the VAF of target detected in WT-cell-free DNA wells. A cell-free DNA sample was called positive if the VAF % of one or both targets analysed was above the false positive rate for the target assay and if the 95% confidence interval error-bars for sample wells did not overlap with the error bars in the control wells as displayed in the concentration plots from QuantaSoft Analysis Pro. Of note, the limit of detection is restricted by the input amount. If only 500 or 1000 haploid genome equivalents of cell-free DNA are available, the theoretical limit of detection would be 0.6% or 0.3% regardless of the false positive rate.

### 2.13. Statistics

For survival analyses, IBM SPSS Statistics version 28.0.0 0 statistic package (IBM Corp., Armonk, NY, USA) was used. All included patients were followed until death, emigration, or end of study period, whichever occurred first. The clinical endpoints examined were overall survival and progression-free survival. Overall survival was defined as the time from gastro-oesophageal cancer diagnosis until death from any cause. Progression-free survival was defined as the time from cancer diagnosis until clinical recurrence or progression was recorded in the medical records (either based on radiology or histopathology/cytology), or death. These curves were presented using the Kaplan–Meier method, and the log-rank test was used to compare groups. Right censoring was applied. Hazard rations (HRs) were calculated with univariable Cox regression. A *p*-value below 0.05 was considered statistically significant, and two-sided *p*-values were reported for all analyses.

## 3. Results

### 3.1. Participants and Tissue Variants

A total of 47 patients were included, and their clinical characteristics can be seen in Table 1 and Appendix A. There are no standard prognostic markers in use clinically for these patients, but all of them receive a tumour stage assessment based on the pathology report for the resected tumour.

Targeted next-generation sequencing using the comprehensive gene panel in combination with manual filtering, with a mean coverage of 1300x, identified potentially cancer-associated variants in 31 of the 47 (66%) pre/intraoperative tissue biopsies. The variants had VAFs of 5–90% (average 25%, median 20%) (Appendix A). The genes with the highest number of variants were *TP53* (25 variants in 24 participants), *KMT2D* (13 variants in 10 participants) and *ARID1A* (12 variants in 11 participants). In 30/31 participants, the detected variants overlapped with the GI cfDNA gene panel, while one individual (P37) had a pathogenic variant in the *ATM* gene in the tissue biopsy, as this gene was not present on the GI cfDNA gene panel. In the remaining 16 participants, no cancer variants with VAF >5% were detected in the tissue analysis. Note that DNA was extracted directly from the small biopsy without histopathological confirmation of a tumour cell fraction.

### 3.2. Detected Diagnostic Plasma Variants

#### 3.2.1. Tumour-Informed Approach

Cancer-associated variants were detected in 31/47 tissue biopsies, and these participants were included in the tumour-informed approach, where 16 out of 31 (52%) participants with cancer-associated pathogenic variants in the tissue analysis had detectable ctDNA in the diagnostic plasma sample, using UMIs to suppress errors (approach A, Figure 3 and Figure 4, Table 2). The detected plasma variants had a median VAF of 0.6% (0.07–17%). A non-UMI-based tumour-informed approach with VarDict (approach B) resulted in the detection of ctDNA in 17 (55%) participants (Figure 3, Table 2).

The genes *TP53* and *ARID1A* harboured the highest number of potentially cancer-associated variants detected in the tumour-informed approaches (15 and 6 variants, respectively). Approach A detected two variants not detected in approach B, and approach B detected five variants not detected by approach A. Fifteen participants had detectable potentially cancer-associated variants in both approaches A and B.

#### 3.2.2. Tumour-Agnostic Approach Using UMIs

Gene panel sequencing was successful in 44 out of 47 diagnostic plasma samples, which could be included in the tumour-agnostic approach (C). In total, 1376 protein-coding variants were detected in the plasma in 44 participants; 82% of them could be excluded by using a blacklist of recurring variants (Appendix A). Additional manual filtering removed 142 variants, resulting in 77 potentially cancer-associated variants in diagnostic plasma from 26 individuals in the cohort (59%). Sixteen of these participants had variants that had been identified in approach A; thus, an additional ten participants with detectable ctDNA were identified using approach C.

Twenty-eight variants were detected in approach C but not present in the tissue analysis cohort and thereby not detectable by the tumour-informed approach (Appendix A). Eight of these variants were cancer hotspot variants for gastrointestinal tumours reported in COSMIC and reported as pathogenic or likely pathogenic in ClinVar. The other 20 variants were all extremely rare in the general population and might be cancer-associated. The variants detected in approach C and also present in the tissue had a median allele frequency of 0.013% (range 0.07–17%), while the variants detected in approach C but not present in the tissue had a median allele frequency of 0.016% (range 0.09–15%).

#### 3.2.3. Removing CH from the Tumour-Agnostic Approach

One potential confounding factor when analysing plasma variants is CH. Variants detected in the plasma that are not tumour-verified (present in tumour tissue DNA) need to be ascertained for CH using either paired DNA from WBC or a repeat plasma sample.

WBC sequencing data were available for 21 individuals in total. These included 17 participants with detectable ctDNA in the tumour-agnostic UMI approach (C), harbouring 62 variants in total. Out of those variants, seven (in the genes *TP53, NF1* and *MYC*) were detected in WBC in >four reads and/or a VAF >0.05% and were thus confirmed to be CH. None of the CH variants were present in the tissue. As a result, one individual was verified as a false positive, with only CH variants (P33) (Appendix A). In total, 55 variants in 16 participants (76%) remained as potential cancer variants after filtering with WBC in approach D. Approach D could thereby identify nine potentially cancer-associated variants in samples whose tissue biopsy had failed. All of the potentially cancer-associated variants detected in both tissue and plasma (approach A) passed the CH filter (Appendix A).

Using the paired plasma samples as CH filtering for 23 additional participants (giving 44 in total), 3 out of the 28 variants detected in approach C but not in the tissue were seen in plasma both before (diagnostic) and after surgery (in the genes *POLE, TP53* and *NF1*). The other 25 variants were not present in the follow-up samples, suggesting that they were not CH.

Approach D had a ctDNA detection rate of 52% (23/44 participants) (Table 2), combining both WBC data and post-surgical sample data. Of note, five of the eight cancer-hotspot variants found by approach C, but not verified in the tissue, were, in fact, CH variants. The *TP53* gene harboured the most CH variants: 4/20 (20%) of all the diagnostic plasma *TP53* variants found in approach C were determined to be CH (Appendix A).

For the planned approach with non-UMI data and WBC variant filtering to remove CH (approach E), it was not possible to determine a robust threshold for the number of reads or VAF (neither in the plasma nor in WBC) for differentiating between CH and cancer-associated variants.

#### 3.2.4. Survival Analyses according to Diagnostic Plasma Samples

Of all participants with cancer stages I–II, 7/25 (28%) had detectable ctDNA in the tumour-informed approach, with a corresponding number of 11/24 (46%, having a denominator of 24, since one sample was coupled to a failed WBC analysis with no post-surgical sample) for the tumour-agnostic approach with CH removal (D). For stages III–IV, the tumour-informed approaches generated a ctDNA detection rate of 9/19 (47%), with the corresponding number for approach D of 12/19 (67%).

Participants with detectable ctDNA using the tumour-informed approach had a shorter overall survival and higher risk for recurrence or progression, and this correlation was significant in the cohort as a whole (Figure 5).

### 3.3. Detected Plasma Variants after Surgery

#### 3.3.1. All Approaches

In total, five participants had detectable ctDNA by approaches A and/or B in their post-surgical sample out of the 28 participants with cancer-associated pathogenic variants in the tissue analysis and an available plasma sample after surgery (Table 2), whereof three also had detectable ctDNA in the diagnostic sample.

The total cell-free DNA concentration was a median of 3.6 times higher in the plasma samples after surgery than the diagnostic samples before surgery, as can be seen in Figure 6. The increase could still be seen 3 months after surgery. There was no evident correlation between CRPmax detected during the inpatient care for the surgical cancer treatment and the change in the total cell-free DNA concentration (Appendix A).

#### 3.3.2. Survival Predictions according to Plasma Samples after Surgery

All five participants with detectable ctDNA in the tumour-informed approaches after surgery had either persistent disease or had a relapse compared to 22 out of 44 (50%) in the nondetectable group. The Kaplan–Meier curves for overall survival in correlation with detectable ctDNA in the plasma samples drawn after surgery are shown in Appendix A. Since two participants, with different survival times, also differed as to whether they had detectable ctDNA or not in approaches A and B, the *p*-values were 0.05 and 0.072, respectively. The same trend was seen for approaches C and D.

### 3.4. ddPCR Approach

In order to validate the panel sequencing, we selected 12 of the analysed participants who had sufficient amounts (≥2 mL) of plasma left for sensitive-targeted ddPCR analysis and one to two cancer-associated SNV/indels in the tissue that could be targeted with wet lab-verified ddPCR assays made by Bio-Rad (Hercules, CA, USA). Nine participants had positive diagnostic plasma samples using the tumour-informed gene panel approach (A and/or B) but negative post-operative samples; two were negative on both the pre- and postoperative samples, and one had a failed diagnostic sample and was negative on the post-operative sample.

In total, 14 recurrent variants, including *TP53* and *KRAS* hotspot mutations, were analysed in plasma using ddPCR mutation assays (Figure 7, Appendix A). The false-positive rate for the ddPCR assays ranged between 0 and 0.06 VAF%. The total number of cell-free DNA copies per sample was a median of 2304 haploid genome equivalents per mL plasma (hGE/mL) (range 560–10,437) in the diagnostic samples and 20,607 hGE/mL (range 1958–542,312) in the post-surgical samples. This confirms the post-operative increase in the total amount of cell-free DNA previously noted through concentration measurements.

The ddPCR could detect its target in 10 out of the included 12 participants (Table 3). Seven of the nine participants with detectable ctDNA by the gene panel and tumour-informed approach also had detection by ddPCR in the diagnostic plasma sample. One of the two participants with a negative pre-operative plasma sample on the gene panel analysis and was positive using ddPCR. Using ddPCR, six participants had detectable post-surgical levels with VAFs 0.01–0.24%, even though none of the 12 participants had detectable ctDNA in the post-surgical samples using the gene panel analysis. Moreover, ddPCR enabled sensitive longitudinal monitoring during neoadjuvant chemotherapy, as well as after surgery (e.g., P12 in Figure 8 and Appendix A).

## 4. Discussion

### 4.1. Summary

In our study of patients with resectable gastro-oesophageal adenocarcinoma, detectable cancer-associated variants in plasma before cancer treatment were associated with shorter overall and progression-free survival than when no variants were detected. We carefully evaluated two different approaches for plasma analysis: tumour-informed and tumour-agnostic, using two different bioinformatic pipelines and with and without the addition of matched normal DNA from WBC or longitudinal plasma sampling. In addition, we also analysed selected participants with ddPCR to further validate the gene panel and compare the sensitivity of the two methods.

In summary, we found tumour-verified cancer-associated variants in 52–55% of our cohort, depending on the bioinformatic pipeline used.

### 4.2. Plasma Panel Design

We designed a custom panel for cell-free DNA analysis of gastro-oesophageal cancer with 30 genes, based on reported variants in gastro-oesophageal cancer in the TCGA/CBioportal database [8]. Based on other pilot tests in our group, the actual number of reads per target region varies depending on the cell-free DNA input and the amount of sequencing. With a small panel, we can obtain at least 500 reads in 95% of the target regions. Using a panel larger than 1 Mb, the same sequencing conditions will only give a target coverage of 100 reads. As we need at least three mutant molecules in the sample in order to call a variant, we will have a theoretical sensitivity of 3/500 = 0.6% with a small panel and 3/100 = 3% with a large panel. Thus, our hypothesis was that we could detect variants of lower VAF if we used a small gene panel, increasing the read count for the targeted regions, even though this comes with the cost of a potential lower biological sensitivity, by restricting the size of the gene panel. We predicted our panel to detect at least 84% of all gastro-oesophageal cancers based on the gene content. After analysis of the tumour sequencing data, there was only one participant that had no possibly detectable ctDNA variant because of the panel design (a pathogenic *ATM* variant not included in the GI cfDNA panel). Therefore, in a future update of the panel, we would consider adding the *ATM* gene.

### 4.3. Plasma Detection Rates

The ratio of participants with detectable ctDNA in the tumour-informed approach A and B was 52–55%. However, only the samples with a paired tissue sample with cancer-associated variants could be included, comprising only 66% of participants and thus limiting the sensitivity of this tumour-informed approach. For all our approaches, we could detect ctDNA variants at lower VAF than 0.5%, the limit of detection stipulated in performance assessment studies carried out using SeraCare reference samples. No cancer variants were detected in the tissue from 16 participants, likely due to low tumour cell fractions in the biopsies, demonstrating the challenge of acquiring representative tissue biopsies and the limitations of a tumour-informed plasma analysis approach. In these cases, cell-free DNA analyses could potentially provide additional diagnostic information without the patient having to be re-biopsied. Our results are comparable to those of Yang et al. They used a tumour-informed, UMI based approach and a 1021-gene panel in 46 patients with gastric cancer stage I-III, and they could show that 45% had a detectable ctDNA in the diagnostic sample. Cancer-associated variants detectable in plasma both before and after surgery were associated with an increased risk of relapse and a shorter overall survival [5]. Azad et al. detected ctDNA in 27/44 (61%) diagnostic plasma samples from localised oesophageal cancer patients, also using a tumour-informed approach [22].

On the contrary, all participants with available diagnostic blood samples, regardless of tissue analysis results, could be included in the tumour-agnostic approach C. Using error-suppression and rigorous filtering, ctDNA was detected in 59% (including all tissue variants also found by the tumour-informed approach). In addition, seven participants, whose tissue biopsy analysis failed, also had detectable ctDNA using this approach. However, nine variants in total were later found to be clonal blood cell variants (CH), see discussion below, thus addition of paired WBC or consecutive sample comparison was necessary for correct interpretation.

Our findings agree with Leal et al., who recently published a tumour-agnostic study, wherein they included 50 treatment-naïve gastric cancer patients and analysed plasma using a panel of 58 cancer driver genes. After exclusion of variants that were present in WBC, they detected likely tumour-specific variants in 54% (27/50) of the participants. They conclude that detection of ctDNA before treatment correlated to a poorer overall survival [4].

Persistent ctDNA detection after surgery was associated to a high risk of relapse and short overall survival. Leal et al. also studied ctDNA after treatment in 20 patients and those with persistent detectable ctDNA had a poorer overall survival. 6/9 with detectable ctDNA after surgery died from metastatic disease [4]. Azad detected ctDNA in post-treatment samples in 5/31 oesophageal cancer patients using a tumour-informed approach, and these patients had an increased risk of progression, distant metastases, and disease-specific death [22].

As we expected, the ddPCR method was the most sensitive, detecting ctDNA in samples with no ctDNA detected by any of the other approaches based on panel sequencing. Similar results were reported by Openshaw et al., who included 40 patients with gastro-oesophageal cancer (22 with curative intent), for a ddPCR study of plasma variants, first defined by tissue analysis. In eight patients, ctDNA could be detected in diagnostic plasma and in five also in post-operative samples. All but one of the patients with detectable ctDNA after surgery relapsed. Thus, the detection of ctDNA after surgery predicted short disease-free survival and the higher the VAFs the poorer the overall survival [23]. ddPCR can provide sensitive monitoring of treatment response and early detection of relapse, which may be of future clinical use. However, this approach requires prior knowledge of the somatic variants in the tumour tissue and custom-designed assays. In addition, even though multiple aberrations are targeted, the mutational landscape may evolve over time and might not match the diagnostic tissue sample during follow-up.

### 4.4. Increasing the Specificity

#### 4.4.1. Sequencing Error Suppression

Using deep sequencing generates thousands of reads over selected target regions, and a proportion of these reads will contain sequencing errors. In a tumour-informed approach, only the known pathogenic variants from the tissue analysis are assayed among all the called variants, and we show that no specific error-suppression approach is needed. However, in tumour-agnostic approaches, the number of variants generated by deep-sequencing will be high, and an error-suppression method is required [24,25]. We used a UMI-based approach in TNscope (Sentieon) to exclude variants that were not present in at least three original molecules. We used a blacklist of artefacts that occurred more than 10 times in our cohort to filter variants and succeeded in removing 80% of all detected variants without losing any of the known tumour-informed variants. Using stringent manual filtering, we could select 77 putative variants in approach C. Nine of these were identified to be CH variants through further analysis of WBC or consecutive plasma samples.

#### 4.4.2. Excluding CH Variants

Clonal haematopoiesis (CH) is the accumulation of clonal expansion of somatic genetic variants in haematopoietic stem cells. It is part of the normal ageing process, and highly prevalent in the general population: up to 50% of all 40 year olds and 75% of all 70-year olds have CH in their white blood cells [26]. CH can also be measured as somatic variants in tumours at low VAFs (<20%) for instance through the presence of tumour-infiltrating lymphocytes or blood cell DNA captured in the tissue analysis [26,27]. In some cases, CH-variants can have a VAF of over 20% in plasma. It has been argued that WBC analysis is necessary to validate that a plasma variant is indeed cancer-associated and not a result of CH. If a tumour-agnostic approach is to be used, matched normal samples or consecutive samples are necessary in order to exclude CH [4,26,28].

There is no consensus cut-off regarding how many reads in WBC analysis would be sufficient for a variant to be considered CH. Different approaches have been suggested, some based on statistical calculations of coverage, or a strict cut off at VAF 0.07 [28,29]. Based on our study, we suggest a cut-off in TNscope of >4 reads, or VAF >0.005 in WBC, for a variant to be considered CH. Of note, none of the variants detected using the tumour-informed approach were found in WBC above our cut-offs, supporting the choice of 5% tissue VAF in order to avoid selecting CH-variants in the tissue analysis.

### 4.5. Plasma Analysis as a Prognostic Biomarker

In our cohort, detectable ctDNA was associated with shorter overall survival and progression-free survival using a tumour-informed approach, but not when using the tumour-agnostic approach. Potentially, CH variants made that approach less sensitive, diluting the number of participants with true cancer-associated variants. Leal et al. could show a significant correlation between overall survival and detection of ctDNA only when removing CH by WBC analysis, in their tumour-agnostic approach [4]. We could see the same trend in our tumour-agnostic approach, though not statistically significant. Varkalaite et al. performed deep sequencing in cell-free DNA of a personalised gene panel in 26 gastric cancer patients and found that the proportion of patients with detectable ctDNA was significantly higher in the sub-cohort with higher tumour burden (T3–T4), compared to lower tumour burden (T1–T2), but no significant difference when comparing the groups with and without distant metastasis. The average survival decreased with increasing number of ctDNA variants [30]. Slagter et al. compared gastric cancer biomarkers CEA and CA 19-9 and ctDNA analysis and showed that both preoperative ctDNA detection and elevated tumour markers were associated to shorter survival. No association was found between ctDNA detection and the other tumour markers, though [31].

After surgery, or any tissue trauma and inflammation, the total concentration of cell-free DNA usually increases and can dilute the fraction of ctDNA [32,33]. Increase in total cell-free DNA concentration has been shown to be associated to a shorter overall survival [34]. There are too few samples with metastatic disease, or cancer-associated SNVs after surgery, in our study to draw any conclusions of whether the tumour stage or total DNA concentration affects the detection rate or not, but it is reasonable to believe increased total concentration of cell-free DNA can dilute the true cancer-associated variants.

## 5. Conclusions

In this study, we evaluated ctDNA detection of SNVs as a prognostic biomarker for gastro-oesophageal cancer. We conclude that, by the tumour-informed approaches, the detection of ctDNA corresponds to shorter overall survival and progression-free survival.

We also concluded that, in a tumour-informed approach, a ctDNA analysis is possible using deep sequencing without a special error suppression protocol. However, the sensitivity is limited by the release of ctDNA into the plasma. More sensitive methods, such as ddPCR, likely increase the detection rate but require personalised assays. The tumour-informed approach requires results from tumour sequencing, which may fail if only small tissue biopsies are used. Therefore, a tumour agnostic approach can detect ctDNA in a larger proportion of patients, but this approach requires artefacts and CH removal, as well as error suppression.

Based on this study and the results from others, we think that clinical implementation of a ctDNA analysis as a prognostic cancer biomarker is possible. A plasma analysis in combination with tumour DNA and WBC sequencing has the potential to improve the management of gastro-oesophageal cancer.

## Figures and Tables

**Figure 1 cancers-15-01160-f001:**
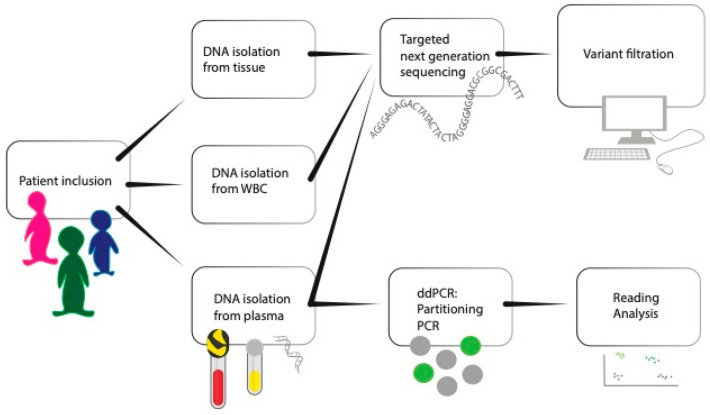
Study workflow. The schematic study plan of the project from patient inclusion to analysis of DNA from the three different materials. WBC, white blood cell; ddPCR, digital droplet PCR, PCR polymerase chain reaction.

**Figure 2 cancers-15-01160-f002:**
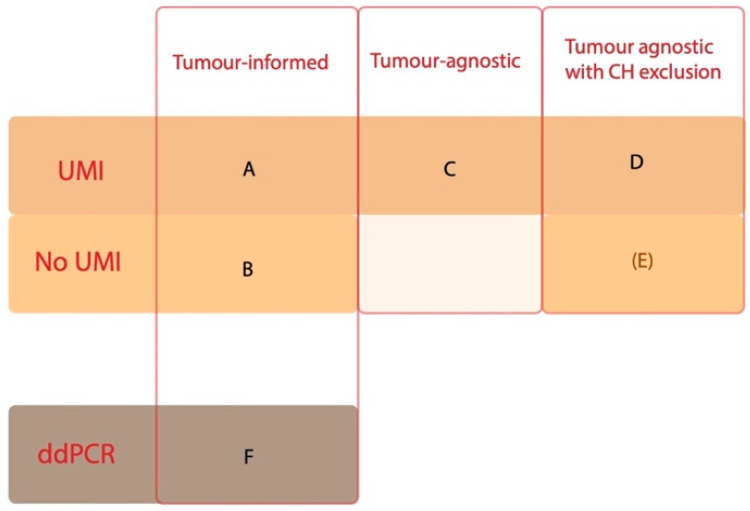
Variant filtering approaches. Three different bioinformatic technical approaches were used (headings on the *x*-axis): tumour-informed (=only tissue variants included), tumour-agnostic (=all filtered plasma variants included) and tumour-agnostic with CH exclusion (=all tissue variants included, except those excluded as CH by the white blood cell analysis or paired sample comparisons). Three different techniques were investigated (headings on the *y*-axis): sequencing with UMIs, without UMIs and ddPCR (digital droplet PCR). Each letter (**A**–**F**) is referred to in detail below. UMI, unique molecular identifiers; ddPCR, digital droplet PCR; CH, clonal haematopoiesis.

**Figure 3 cancers-15-01160-f003:**
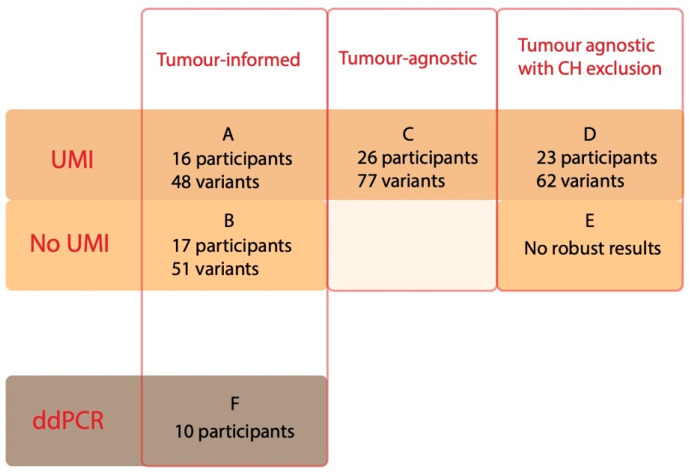
Variants and number of participants with detectable variants for each filtering approach in the diagnostic plasma sample. The number of participants with detectable ctDNA, and the total number of detected variants for each approach is presented in each box, sorted by the bioinformatics approach. In the tumour-informed approach (**A**,**B**), there were expected ctDNA findings in 31 participants (98 variants) based on findings in the tissue analysis and an available diagnostic plasma sample. In the tumour-agnostic approach (**C**), there were plasma samples available for 44 participants (making them the total number of participants with expected ctDNA findings). In the tumour-agnostic approach with CH exclusion (**D**,**E**), there were white blood cells and/or two plasma samples available for 43 participants (making them the total number of participants with expected findings). For the ddPCR approach, which was tumour-informed (**F**), 12 participants who were also included in approaches A and B with 19 variants were selected (making them the total number of participants with expected ctDNA findings). ctDNA, cell-free tumour DNA; UMI, unique molecular identifiers; ddPCR, digital droplet PCR; CH, clonal haematopoiesis.

**Figure 4 cancers-15-01160-f004:**
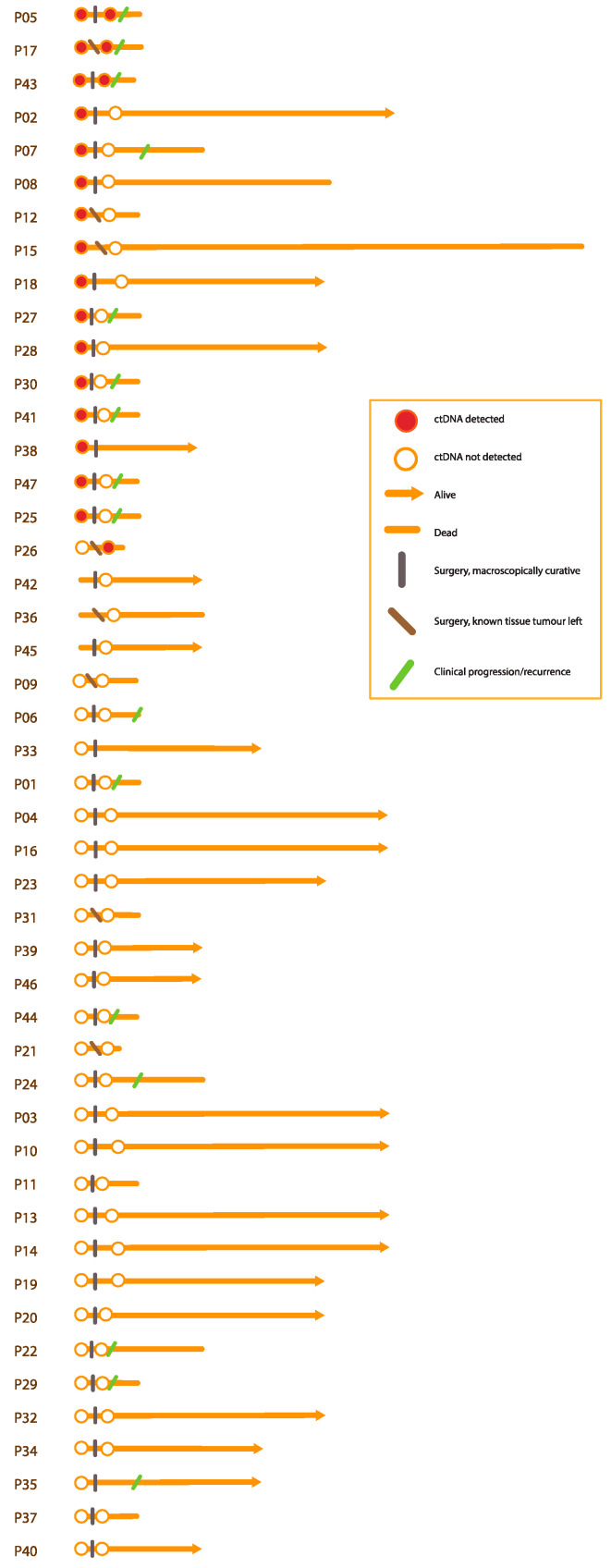
ctDNA detection and clinical course for each participant. Each participant is represented by an orange lifeline, starting with the diagnostic plasma sample. The *x*-axis represents time from diagnosis. The detection status of ctDNA for each participant is represented by a circle, and participants still alive at the end of the study have a lifeline ending with an arrow. Time of surgery and recurrence are marked by vertical lines, as shown in the orange box on the right. For the specific variants detected and their variant allele frequencies, please refer to Appendix A. ctDNA, cell-free tumour DNA.

**Figure 5 cancers-15-01160-f005:**
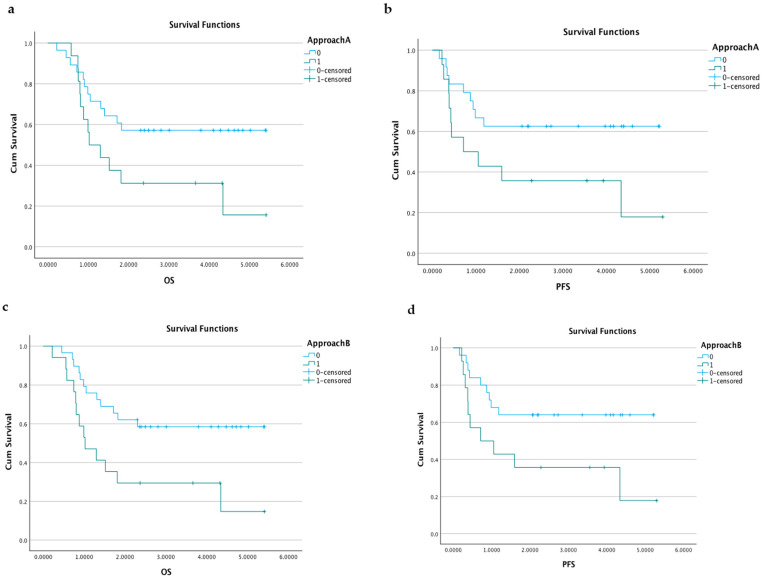
Kaplan–Meier curves. Kaplan–Meier curves of overall survival and progression-free survival, in correlation with detectable ctDNA (1, green curve) and no detectable ctDNA (0, blue curve) in the diagnostic plasma sample: (**a**) Kaplan–Meier curve of the overall survival based on ctDNA detection using approach A (denominated according to Figure 2) for the diagnostic sample. Log rank *p* = 0.048. (**b**) Kaplan–Meier curve of progression-free survival based on ctDNA detection using approach A (denominated according to Figure 2) for the diagnostic sample. Log rank *p* = 0.049. (**c**) Kaplan–Meier curve of the overall survival based on ctDNA detection using approach B (denominated according to Figure 2) for the diagnostic sample. Log rank *p* = 0.011. (**d**) Kaplan–Meier curve of progression-free survival based on ctDNA detection using approach B (denominated according to Figure 2) for the diagnostic sample. Log rank *p* = 0.03. The survival times and censoring are also specified in Appendix A. In approaches C and D, a similar tendency was seen, but there was no significant difference in survival between the participants with detectable ctDNA and those without. ctDNA, cell-free tumour DNA; OS, overall survival; PFS, progression-free survival.

**Figure 6 cancers-15-01160-f006:**
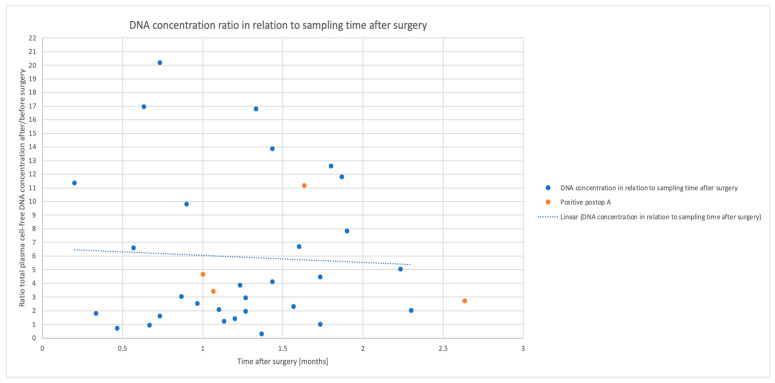
Cell-free DNA concentration ratio in relation to the sampling time after surgery. The ratio between the diagnostic plasma sample total cell-free DNA concentration and the concentration within 3 months after surgery is shown on the *y*-axis, with time after surgery on the *x*-axis. The blue horizontal line represents ratio 1, i.e., no difference in concentration. Most samples had an increased total cell-free DNA concentration (above ratio 1 on the *y* axis) between the two samples. Positive postop A, ctDNA detected after surgery by the bioinformatic approach A (tumour-informed).

**Figure 7 cancers-15-01160-f007:**
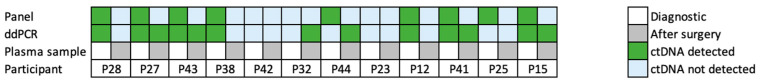
Heatmap of ddPCR results in relation to gene panel results. The ctDNA detection with targeted panel sequencing is compared to detection with ddPCR of one or two selected target variants. Green represents detection of ctDNA with the specific method, and light blue represents no detection of ctDNA. Both the diagnostic plasma sample (white box) and the sample from after surgery (grey box) are shown. For most samples, ddPCR is the most sensitive method. ddPRC, digital droplet PCR; PCR, polymerase chain reaction; ctDNA, cell-free tumour DNA.

**Figure 8 cancers-15-01160-f008:**
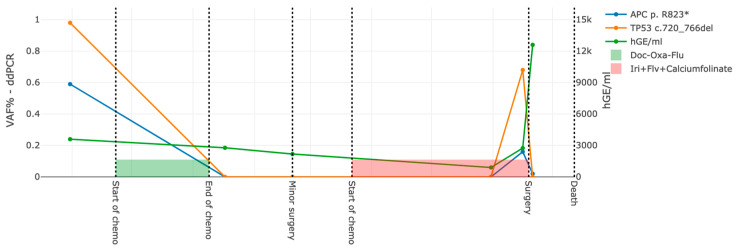
ddPCR detection in longitudinal plasma sampling. One example (P12) of ddPCR results for dual ctDNA variant targets in relation to surgery and chemotherapy. On the *y*-axis, levels of detected *APC* and *TP53* pathogenic variants in cell-free DNA by ddPCR, as well as the total amount of cell-free DNA (hGe/mL). hGE/mL is defined by the scale on the right *y*-axis. On the *x*-axis is the time from diagnosis. The patient received neoadjuvant chemotherapy (the combination Docetaxel–Oxaliplatin–Fluorouracil and then Docetaxel–Oxaliplatin–Calcium Folinate), and CT (computer tomography) showed a decreased size of the main tumour and lymph node metastases. A diagnostic laparoscopy (named “minor surgery”) was performed after the chemotherapy, which revealed suspected abdominal carcinosis. The patient therefore received a second round of chemotherapy. CT showed decreased size of the lymph node metastases, and the patient underwent major surgery (marked as “surgery”). After the main surgery, the patient suffered from pneumonia, bleeding from the anastomosis and skin infection. This led to sepsis and death soon after the surgery. Graphs for all 12 participants can be seen in Appendix A, and the 2D and 3D plots in Appendix A. ddPCR, digital droplet PCR; hGe, human genome equivalents. * truncating.

**Table 1 cancers-15-01160-t001:** Demography and tumour information for all 47 included participants.

Participant ID	Cancer Localisation	Age Diagnosis	Sex	Tumour Stage (Based on Pathology Report *)	Overall Survival (Years)	Alive at the End of Study
P01	Stomach	71	Man	IIIC	1.31	no
P02	Stomach	84	Man	IIB	5.41	yes
P03	Stomach	79	Man	IIB	5.41	yes
P04	Stomach	79	Man	IB	5.39	yes
P05	Stomach	78	Man	IIIC	0.75	no
P06	Stomach	78	Man	IIIC	0.90	no
P07	Stomach	74	Man	IIA	1.81	no
P08	Stomach	64	Man	IB	4.34	no
P09	Stomach	59	Woman	IIIC	1.40	no
P10	Stomach	55	Man	IB	5.02	yes
P11	Oesophagus	75	Man	IIIC	0.99	no
P12	Stomach	65	Woman	IIIC-IV	0.81	no
P13	Stomach	75	Woman	IA	4.62	yes
P14	Stomach	57	Woman	IA	4.84	yes
P15	Stomach	74	Man	IIIC	0.79	no
P16	Stomach	68	Woman	IB	4.72	yes
P17	Oesophagus	56	Woman	IV	0.99	no
P18	Stomach	71	Man	IIA	4.32	yes
P19	Stomach	77	Woman	IIB	4.28	yes
P20	Stomach	43	Man	IIB	4.48	yes
P21	Stomach	76	Man	IIIC	0.22	no
P22	Oesophagus	48	Man	IIIC	1.83	no
P23	Stomach	75	Man	IIA	4.11	yes
P24	Stomach	85	Man	IIIC	1.71	no
P25	Stomach	76	Man	IIIC	0.74	no
P26	Stomach	68	Man	IV	0.45	no
P27	Stomach	81	Man	IIIB	0.88	no
P28	Oesophagus	81	Woman	IIIC	3.66	yes
P29	Stomach	74	Woman	IIIC	1.05	no
P30	Oesophagus	67	Man	IIIC	1.02	no
P31	Oesophagus	74	Woman	IIIC	0.88	no
P32	Stomach	80	Man	IA	3.79	yes
P33	Oesophagus	77	Man	IIIA	2.81	yes
P34	Stomach	36	Woman	IIIC	3.01	yes
P35	Stomach	64	Man	IIIB	2.63	yes
P36	Stomach	62	Woman	IV	2.31	no
P37	Stomach	81	Woman	IB	0.71	no
P38	Stomach	77	Woman	IIA	2.37	yes
P39	Oesophagus	61	Man	IIA	2.49	yes
P40	Stomach	72	Man	IB	2.30	yes
P41	Stomach	87	Man	IIIC	0.57	no
P42	Stomach	61	Man	IA	2.31	yes
P43	Stomach	79	Man	IIIC	1.52	no
P44	Oesophagus	83	Woman	IIIA	0.55	no
P45	Oesophagus	72	Man	IA	2.36	yes
P46	Stomach	64	Man	IB	2.39	yes
P47	Stomach	82	Woman	IIIA	1.30	no

Table 1 Legend: The table states clinical data from each participant. Further clinical details can be seen in Appendix A. * As reported by the Pathology Department, Karolinska University Hospital, Stockholm, Sweden. Analysis performed on clinical tumour tissue samples.

**Table 2 cancers-15-01160-t002:** Number of ctDNA variants per approach, and participants and the final amounts of plasma and DNA used.

Participant ID	Expected Variants *	A. Diagnostic	B. Diagnostic	C. Diagnostic	D. Diagnostic	A. Post	B. Post	C. Post	D. Post	Plasma Diagnostic (ng)	Plasma Diagnostic (mL)	Plasma Post (ng)	Plasma Post (mL)
P01	N	0	0	1	1	0	0	0	0	36.9	5	21.2	3
P02	Y	1	1	2	1	0	0	0	0	17.9	3	8.5	2
P03	Y	0	0	0	0	0	0	0	0	26.3	4	28.3	3
P04	N	0	0	4	4	0	0	0	0	42.9	4	54.5	2
P05	Y	9	9	10	7	4	4	4	1	73.5	5	206	3
P06	N	0	0	0	0	0	0	1	1	250	5	75	3
P07	Y	1	1	2	2	0	0	0	0	33.3	3	60	3
P08	Y	2	2	4	4	0	0	0	0	26.3	3	81	4
P09	N	0	0	1	1	0	0	0	0	68	3	69	5
P10	Y	0	0	0	0	0	0	1	1	25.5	5	32.2	3
P11	Y	0	0	0	0	0	0	0	0	45.9	2	139	3
P12	Y	3	3	4	4	0	0	0	0	81.5	3	247	3
P13	N	0	0	0	0	0	0	0	0	66	5	250	4
P14	N	0	0	0	0	0	0	0	0	110	5	250	3
P15	Y	2	2	2	2	0	0	0	0	236	5	250	3
P16	N	0	0	2	1	0	0	1	0	39.3	5	231	3
P17	Y	3	3	4	3	2	2	2	2	26.7	4	68.5	3
P18	Y	15	15	15	15	0	0	0	0	17.6	4	28.4	4
P19	Y	0	0	0	0	0	0	0	0	189	5	47	4
P20	N	0	0	0	0	0	0	0	0	47	5	250	3
P21	Y	0	1	0	0	0	0	0	0	55.5	5	250	3
P22	Y	0	0	0	0	0	0	0	0	39.2	5	29	3
P23	Y	0	0	1	1	0	0	0	0	20.8	4	250	3
P24	Y	0	0	0	0	0	1	0	0	134	5	158	3
P25	Y	1	0	1	1	0	0	0	0	48.6	4	250	3
P26	Y	0	0	0	0	1	0	1	1	16.8	4	141	3
P27	Y	1	1	3	1	0	0	0	0	222	5	250	4
P28	Y	3	3	3	3	0	0	0	0	52.5	3	155	3
P29	Y	0	0	0	0	0	0	0	0	106	3	197	3
P30	Y	2	2	3	3	0	0	0	0	19.2	4	72.5	3
P31	N	0	0	1	1	0	0	0	0	24.6	3	250	5
P32	Y	0	0	0	0	0	0	0	0	71.5	4	250	4
P33	Y	0	0	2	0	NA	NA	NA	NA	116	4	NA	4
P34	N	0	0	0	0	0	0	0	0	35.8	4	120	3
P35	N	0	0	0	0	0	0	0	0	96	4	149	4
P36	N	NA	NA	NA	NA	0	0	0	NA	NA	4	241	4
P37	Y **	0	0	0	0	0	0	0	0	65	4	250	4
P38	Y	1	1	2	NA	NA	NA	NA	NA	24	4	NA	4
P39	N	0	0	1	1	0	0	0	0	11.1	3	168	4
P40	Y	0	0	0	0	0	0	0	0	38.7	4	250	4
P41	Y	1	1	2	2	0	0	0	0	47	4	250	4
P42	Y	NA	NA	NA	NA	0	0	0	NA	NA	4	250	4
P43	Y	3	4	4	2	1	1	2	0	98.5	4	250	4
P44	Y	0	1	1	0	0	0	1	0	125	4	250	4
P45	N	NA	NA	NA	NA	0	0	0	NA	NA	NA	55	3
P46	N	0	0	1	1	0	0	0	0	33	3	250	4
P47	Y	1	1	1	1	0	0	0	0	51.5	4	250	4

Table 2 Legend: ctDNA detection results for each participant and each bioinformatic approach (same denomination as stated in Figure 2A–D), and the amount of plasma used for the two different sample analyses per participant. Diagnostic, plasma sampled before any treatment; Post, plasma sampled after surgery; TI, tumour-informed; TA, tumour-agnostic; A, B, C and D, bioinformatic approaches based on the definitions in Figure 2. * Cancer-associated variants detected in the tissue analysis; ** gene not included in the plasma panel. N, no; Y, yes; NA, not applicable; ctDNA, cell-free tumour DNA.

**Table 3 cancers-15-01160-t003:** Longitudinal cell-free DNA analysis using ddPCR, including input volumes.

Participant	Sample Type and plasma Volume	Target	False-Positive Rate	Total Cell-Free DNA Copies/mL	VAF [%] ddPCR	VAF [%] Gene Panel	ddPCR Result per Variant	ddPCR Result per Patient	Gene Panel Results Tumour-Informed Approaches
P28	Diagnostic, 3 mL	*TP53* p. R273H	0.03	1051	0.5	0.4	positive	positive	positive
		*KRAS* p. G13D	0.03		0.5	0.3	positive		positive
	Post-surgery, 3 mL	*TP53* p. R273H	0.03	1958	0	0	negative	negative	negative
		*KRAS* p. G13D	0.03		0.03 **	0	negative		negative
	Follow-up-1, 3 mL	*TP53* p. R273H	0.03	456	0	NA	negative	negative	NA
		*KRAS* p. G13D	0.03		0	NA	negative		NA
	Follow-up-2, 5 mL	*TP53* p. R273H	0.03	1002	0	NA	negative	negative	NA
		*KRAS* p. G13D	0.03		0	NA	negative		NA
P27	Diagnostic, 5 mL	*PIK3CA* p. H1047R	0	8151	0.01	0	positive	positive	negative
		*TP53* p. R273C	0.05		1.32	0.6	positive		positive
	Post-surgery, 3 mL	*PIK3CA* p. H1047R	0	27,247	0.006	0	negative	positive	negative
		*TP53* p. R273C	0.05		0.1	0	positive		negative
	Follow-up-1, 3 mL	*PIK3CA* p. H1047R	0	1156	0	NA	negative	negative	NA
		*TP53* p. R273C	0.05		0	NA	negative		NA
P43	Diagnostic, 4 mL	*KRAS* p. G12D	0.06	2154	0.32	0.5	positive	positive	positive
	Post-surgery, 3 mL	*KRAS* p. G12D	0.06	5348	0.24	0	positive	positive	negative
	Follow-up-1, 5 mL	*KRAS* p. G12D	0.06	2183	0.6	NA	positive	positive	NA
P38	Diagnostic, 4 mL	*TP53* p. R248W	0.01	1119	0.19	0.3	positive	positive	positive
	Post-surgery, 5 mL	*TP53* p. R248W	0.01	7111	0,01	0	negative	negative	negative
P41	Diagnostic, 4 mL	*RB1* p. R358 *	0.04	933	14.7	0	positive	positive	negative
		*TP53* p. R273C	0.03		6.8	17	positive		positive
	Post-surgery, 4 mL	*RB1* p. R358 *	0.04	28,862	0.17	0	positive	positive	negative
		*TP53* p. R273C	0.03		0.27	0	positive		negative
	Follow-up-1, 4 mL	*RB1* p. R358 *	0.04	1376	3.15	NA	positive	positive	NA
		*TP53* p. R273C	0.03		1.42	NA	positive		NA
P25	Diagnostic, 4 mL	*TP53* p. R196 *, c.586C>T	0.02	5080	0.02	0.07	negative	negative	positive
	Post-surgery, 3 mL	*TP53* p. R196 *, c.586C>T	0.02	131,322	0.0003	0	negative	negative	negative
	Follow-up-1, 3 mL	*TP53* p. R196 *, c.586C>T	0.02	1898	0.02	NA	negative	negative	NA
	Follow-up-2, 3 mL	*TP53* p. R196 *, c.586C>T	0.02	2928	0.3	NA	positive	positive	NA
P12	Diagnostic, 3 mL	*APC* p. R823 *	0.04	3597	0.59	0,7	positive	positive	positive
		*TP53* c.720_766del	0		0.98	0,3	positive		positive
	Chemo-follow1, 3 mL	*APC* p. R823 *	0.04	2776	0	NA	negative	negative	NA
		*TP53* c.720_766del	0		0	NA	negative		NA
	Chemo-follow2, 3 mL	*APC* p. R823 *	0.04	2180	0	NA	negative	negative	NA
		*TP53* c.720_766del	0		0	NA	negative		NA
	Chemo-follow3, 3 mL	*APC* p. R823 *	0.04	897	0	NA	negative	negative	NA
		*TP53* c.720_766del	0		0	NA	negative		NA
	Chemo-follow4, 3 mL	*APC* p. R823 *	0.04	2748	0.16	NA	positive	positive	NA
		*TP53* c.720_766del	0		0.68	NA	positive		NA
	Post-surgery, 3 mL	*APC* p. R823 *	0.04	12,596	0.02	0	negative	negative	negative
		*TP53* c.720_766del	0		0	0	negative	negative	negative
P15	Diagnostic, 3 mL	*TP53* p. R248Q, c.743G>A	0.04	10,437	0.27	0,23	positive	positive	positive
		*ERBB2* p. S310Y, c.929C>A	0		0.19	0	positive		negative
	Post-surgery, 3 mL	*TP53* p. R248Q, c.743G>A	0.04	13,968	0.04	0	negative	positive	negative
		*ERBB2* p. S310Y, c.929C>A	0		0,01	0	positive		negative
	Follow-up1, 3 mL	*TP53* p. R248Q, c.743G>A	0.04	7323	3.71	NA	positive	positive	NA
		*ERBB2* p. S310Y, c.929C>A	0		4.0	NA	positive		NA
P44	Diagnostic, 3 mL	*TP53* p. R273H	0.03	576	0.25 **	0,26	negative	negative	positive
		*KRAS* p. G13D	0.03		0	0	negative		negative
	Post-surgery, 4 mL	*TP53* p. R273H	0.03	4879	0.1	0	positive	positive	negative
		*KRAS* p. G13D	0.03		0.07	0	positive		negative
P42	Diagnostic, 4 mL	*TP53* p. R158fs *12, c.472del	0.02	5721	0.006	0	negative	negative	NA
	Post-surgery, 4 mL	*TP53* p. R158fs *12, c.472del	0.02	38,515	0	0	negative	negative	negative
P32	Diagnostic, 4 mL	*TP53* p. R248W	0.02	2455	0	0	negative	negative	negative
	Post-surgery, 5 mL	*TP53* p. R248W	0.02	100,726	0.04	0	positive	positive	negative
	Follow-up-1, 3 mL	*TP53* p. R248W	0.02	1227	0	NA	negative	negative	NA
	Follow-up-2, 5 mL	*TP53* p. R248W	0.02	1944	0	NA	negative	negative	NA
P23	Diagnostic, 4 mL	*TP53* p. R273H	0.03	560	0	0	negative	negative	negative
		*KRAS* p. G13D	0.03		0.2 **	0	negative		negative
	Pre-surgery-2, 4 mL	*TP53* p. R273H	0.03	1054	0	NA	negative	negative	NA
		*KRAS* p. G13D	0.03		0	NA	negative		NA
	Post-surgery, 3 mL	*TP53* p. R273H	0.03	542,312	0	0	negative	negative	negative
		*KRAS* p. G13D	0.03		0	0	negative		negative
	Follow-up-1, 3 mL	*TP53* p. R273H	0.03	8043	0.01	NA	negative	negative	NA
		*KRAS* p. G13D	0.03		0.03	NA	negative		NA
	Follow-up-2, 3 mL	*TP53* p. R273H	0.03	766	0	NA	negative	negative	NA
		*KRAS* p. G13D	0.03		0	NA	negative		NA

Table 3 Legend: Detection results from ddPCR for each plasma sample in 12 participants for each target assay. When ctDNA was detected, it is considered positive. The last column includes the results from the cell-free DNA analysis by gene panel sequencing. The volumes of plasma used for each reaction are presented in the second column. * truncating, ** Determined negative because of overlapping error bars with the negative control. The false-positive rate was determined by the VAF (%) in the negative control plasma (9–12 wells). Diagnostic, cell-free DNA sampled before start of treatment; post-surgery, sampled after surgery; follow-up, sampled during the clinical follow-up period (can be multiple samples, increasing numbers according to time from diagnosis); ddPCR, digital droplet PCR; VAF, variant allele frequency; NA, not applicable.

## Data Availability

Data are freely available after individual application via SciLife Lab Data Resposity (http://doi.org/10.17044/scilifelab.22067456, accessed on 7 February 2022).

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
