# Peer review of "Sensitive Detection of Cell-Free Tumour DNA Using Optimised Targeted Sequencing Can Predict Prognosis in Gastro-Oesophageal Cancer"

_cancers, 2023, doi:10.3390/cancers15041160_

Round 1

Reviewer 1 Report

The work is very interesting, however, a few issues need to be completed:

1) the aim of the work is: "In the present study, we aim to analyze single-nucleotide variants (SNVs) in ctDNA from plasma as a prognostic biomarker." and final conclusions should answer this question, while the conclusions focus on the not fully known effect of the total concentration of free circulating DNA before and after the surgery. These issues should be structured and clearly presented.

2) in the case of assessing the prognostic value of a marker, the work should be prepared in accordance with the REMARK reccomendation

3) the manuscript, and not just supplementary materials, should include a complete list of the gene panel as well as digital PCR assays

4) the description of the methodology should contain relevant information on the digital PCR assays used, along with information on multiplex reactions and what they contained precisely; paragraph 2.5 does not contain any description, including the amount of plasma taken for testing and the concentrations obtained for individual patients,  repeated phrases from the leading and supplementary text should be removed

5) the description of the obtained results in section 3.3 Detected plasma variants after surgery contains mainly information about the cfDNA concentration

6) figures presented in a file described as Supplemental Table S6 should have legends explaining what is presented on them

7) the nomenclature of gene names should be checked in the attached supplementary materials

Author Response

We would like to thank the reviewer for the careful and thorough revision, and the improvement suggestions and comments. We address each specific issue below. All corrections and additions to the original manuscript are marked in yellow.

  • the aim of the work is: "In the present study, we aim to analyze single-nucleotide variants (SNVs) in ctDNA from plasma as a prognostic biomarker." and final conclusions should answer this question, while the conclusions focus on the not fully known effect of the total concentration of free circulating DNA before and after the surgery. These issues should be structured and clearly presented.

We thank the reviewer for pointing this out. In the Conclusion paragraph on lines 526-541, we have made the concluding remarks more clearly connected to the aim.

  • in the case of assessing the prognostic value of a marker, the work should be prepared in accordance with the REMARK reccomendation

We are thankful for this valuable input and agree this should be made more clear. We have used the REMARK checklist, as presented by Sauerbrei et al 2018 (doi 10.1093/jnci/djy088) and adjusted in the text where clarifications were needed, as described below.

Introduction:
We have added information about the pre-specified hypothesis on line 67-70.

Materials and Methods:
We have added information on lines 76-77, 128-129, 172-173, about matching, blinding of endpoints, and clearly stating the clinical endpoints.

We have also added a REMARK diagram of patient flow (Figure S1), now referred to on line 93-94. (This also meant we had to change the numbering of the other supplementary figures.)

Results:
We have added information on lines 185-187, to address the issue of potential comparable biomarkers in clinical use.

  • the manuscript, and not just supplementary materials, should include a complete list of the gene panel as well as digital PCR assays

We have added a list of all genes in the GI cfDNA panel in the main text on lines 116-120. We also made the description of the GMCK gene panel more clear, on lines 129-136 in the Supplementary Methods. We added the ddPCR assays on lines 165-168 in the main manuscript.

  • the description of the methodology should contain relevant information on the digital PCR assays used, along with information on multiplex reactions and what they contained precisely; paragraph 2.5 does not contain any description, including the amount of plasma taken for testing and the concentrations obtained for individual patients,  repeated phrases from the leading and supplementary text should be removed

We have adjusted the phrases so that they are not repeated in the Supplementary Methods section (lines 237-241 in the Supplementary Methods document, and in paragraph 2.5 in the main document). We have also added more information about the ddPCR assays in that paragraph.

  • the description of the obtained results in section 3.3 Detected plasma variants after surgery contains mainly information about the cfDNA concentration

We agree that the focus should be ctDNA and not cfDNA, and therefore we have moved the paragraphs so that the initial one is about the detection of ctDNA.

  • figures presented in a file described as Supplemental Table S6 should have legends explaining what is presented on them

We have now moved all explanatory comments to a legend above the Table S6, see revised version.

  • the nomenclature of gene names should be checked in the attached supplementary materials

We have run all the gene names in HGNC (https://www.genenames.org/tools/multi-symbol-checker/). All were correct, except MLL/KMT2C which should not have a double name and this has been revised in Table S1 to KMT2C only.

Reviewer 2 Report

The paper titled "Sensitive detection of cell-free tumor DNA using optimized targeted sequencing can predict prognosis in gastro-esophageal cancer" is well written and shows that targeted sequencing could be useful in the clinic for prognosis. There are 47 patients where some have longitudinal follow up to understand the change in ctDNA with surgery. This study uses a small set of designed genes to detect cancer with low variant frequency. Overall the data support the conclusions and set a baseline for future research.

Comments:

In table 2, there is a green box at the bottom Y, I assume this has no meaning and can be removed

Were there any instances of CH and tumor tissue biopsy DNA variants matching? If so, what is the cause (two independent mutations or WBCs in the tissue biopsy)

For the survival predictions after surgery, are these differences (4/4 vs 22/44) statistically significant, or is the sample size too small?

For the longitudinal analysis of P12 (Figure 8) did the spike in TP53 DNA correspond with tumor progression (and thus the need for surgery) or was surgery already planned and the spike did not correlate with anything clinically.

In the discussion on plasma panel design, can you discuss a little more about how much more sensitive your assay is compared to deep sequencing alone?

Author Response

First of all, we would like to thank the reviewer for well-put and well thought through comments and suggestions. We address each specific comment below.

In table 2, there is a green box at the bottom Y, I assume this has no meaning and can be removed

We have removed the green box (and have reformatted all tables in word format).

Were there any instances of CH and tumor tissue biopsy DNA variants matching? If so, what is the cause (two independent mutations or WBCs in the tissue biopsy)

We used a VAF cut-off of 5% in order to call a variant in the tumour tissue NGS-analysis. As we stated on line 261, we found no variants in WBC that we also detected in the tissue analysis, which suggests that our cut-off of 5% seems to be sufficient to remove CH variants.

For the survival predictions after surgery, are these differences (4/4 vs 22/44) statistically significant, or is the sample size too small?

We initially chose not to include the survival graphs for the post-surgical samples, as we agree the sample sizes are small. However, as the reviewer points out, this is of interest despite the small cohort. We have now included the Kaplan-Meier graphs for overall survival as a supplementary figure (S3), and added a reference to it on lines 342-346.

For the longitudinal analysis of P12 (Figure 8) did the spike in TP53 DNA correspond with tumor progression (and thus the need for surgery) or was surgery already planned and the spike did not correlate with anything clinically.

The surgery was already planned, based on the chemotherapy period being finished. We have added clinical information in the legend to Figure 8, and also added information into the figure (minor surgery and time of death).

In the discussion on plasma panel design, can you discuss a little more about how much more sensitive your assay is compared to deep sequencing alone?

A small cell-free DNA panel enables deep sequencing with a target coverage of at least 500 reads in 95% of the regions. A larger panel (>1Mb) will obtain around 100 reads in 95% of the target regions using the same sequencing conditions. As we need at least three mutant molecules in the sample in order to call a variant, we will have a theoretical sensitivity of 3/500 = 0.6% with a small panel and 3/100 = 3% with a large panel.  We have added a few lines on this in the discussion on page 14 (lines 399-405).

Round 2

Reviewer 1 Report

After reviewing the amendments, I still feel that many aspects of the work are unclear:

1) it is not specified by which method nucleic acids were isolated from individual tissues, what volume of plasma or whole blood or tissue for isolation was taken in individual patients; very general information was given, i.e. 2-5 ml - does this mean that 2 ml from one patient and 5 ml from another, etc. there is no description of the standardization of these stages; I will ask for detailed information on each tissue for each patient

2) Figure S1 completely excludes samples that have been digitally PCRed

3) I am asking for precise information in the case of a complete set of analyses, i.e. tissue, WBC and plasma with both techniques and their results

4) please also refer to why digital was not performed in all samples and why it was performed at all, since there is nothing in the conclusions about this analysis

5) The authors use the terms ctDNA and cfDNA interchangeably throughout the manuscript. I want to emphasize that cfDNA also contains non-cancerous DNA, which is important for plasma conclusions. In addition, there is the question of the range of ml taken for isolation. I would like to note that the larger the volume, the greater the chance of obtaining a higher concentration, so the question of what volumes have been used for individual patients is crucial.

6) please attach digital PCR figures presenting results to the manuscript and describe precisely what they show.

7) what was the control for Digital PCR experiments.

Author Response

We thank the reviewer for addressing the issues below, and for helping us to improve the manuscript. We had originally prepared the manuscript with the detailed materials and methods in the Supplementary methods, but have now moved the entire M&M into the main manuscript as suggested by the reviewer.

  • it is not specified by which method nucleic acids were isolated from individual tissues, what volume of plasma or whole blood or tissue for isolation was taken in individual patients; very general information was given, i.e. 2-5 ml - does this mean that 2 ml from one patient and 5 ml from another, etc. there is no description of the standardization of these stages; I will ask for detailed information on each tissue for each patient

This information is now easily accessible in the main manuscript (see sections 2.3 – 2.5). When it comes to specific volumes and amounts, we have added this information in Table 2 and Table 3 for plasma and on line 118-119 for tissue.

  • Figure S1 completely excludes samples that have been digitally PCRed

Figure S1 includes all participants in the ddPCR substudy. The participants for ddPCR were selected from the main cohort. We have now clarified this by adding additional flow chart boxes for the methods used.

  • I am asking for precise information in the case of a complete set of analyses, i.e. tissue, WBC and plasma with both techniques and their results

As mentioned above under point 1, we have now moved all method information from the supplementary section into the main manuscript. We have added an extra table (Table S3) on the ddPCR assays and PCR conditions, which we missed to include in the initial submission. We have also moved the ddPCR result table from the supplementary to the main manuscript (now Table 3). We have added complementary information on the volumes of plasma and amounts of cfDNA used to Table 2. The specific information on the variants detected in tissue can be found in Table S5 and those found in plasma in Table S7 as these tables are too large to move to the main text.

  • please also refer to why digital was not performed in all samples and why it was performed at all, since there is nothing in the conclusions about this analysis

We thank the reviewer for pointing out that this was not clear. We have clarified the selection of participants for ddPCR in section 3.4, and added discussions and conclusions concerning the ddPCR on lines 643-645, 714-796, and 786-787.

In short, ddPCR requires prior knowledge of tumour-specific aberrations, and there had to be enough plasma left (at least 2ml) after panel sequencing to perform the ddPCR reactions. In addition, since we used ddPCR as a supplementary method, we chose to only target SNVs/indels with a wet-lab verified assay from BioRad. Thereby, we could include in this substudy, 12 participants, representing both individuals with detectable and not detectable ctDNA in the gene panel approaches.

  • The authors use the terms ctDNA and cfDNA interchangeably throughout the manuscript. I want to emphasize that cfDNA also contains non-cancerous DNA, which is important for plasma conclusions. In addition, there is the question of the range of ml taken for isolation. I would like to note that the larger the volume, the greater the chance of obtaining a higher concentration, so the question of what volumes have been used for individual patients is crucial.

We agree with this, and to be more clear about when we refer to the total cell-free DNA, as opposed to the specific cancer-associated ctDNA, we have changed into always writing “cell-free DNA” instead of “cfDNA”. Also, we have changed the title of Table 3. Now “cfDNA” is only used when referring to the GI cfDNA gene panel, since that is its name.

As noted under point 1 above, we have now added the individual volumes and amounts.

  • please attach digital PCR figures presenting results to the manuscript and describe precisely what they show

To complement the longitudinal graphs with the ddPCR results presented in Figure S5, we have added Figure S6 with the 1D and 2D plots and concentration graphs from Quantasoft. Both figures have explanatory figure legends.  

  • what was the control for Digital PCR experiments.

This information is in the text, please refer to lines 369-374.

Round 3

Reviewer 1 Report

The authors have introduced most of the details necessary for the clarity of the experiments performed, but the manuscript is still poorly prepared. In the tables, figures and descriptions in the legends, there are no explanations of the abbreviations used ( e.g OS), descriptions of the axes and explanations of what different colors, shapes, etc. mean. All these details should be completed so that even the reader who is not familiar with these methods does not wonder what the figures, tables and graphs present.

Author Response

We thank the reviewer for the comment and improvement suggestion. All figures, tables and diagrams now have legends with the abbreviations spelled out, and we have added descriptive information to the legends for most illustrations. Please see text and cells marked with yellow in the manuscript and all supplementary tables and figures.